# Assessment of a Coated Mitomycin-Releasing Biodegradable Ureteral Stent as an Adjuvant Therapy in Upper Urothelial Carcinoma: A Comparative In Vitro Study

**DOI:** 10.3390/polym14153059

**Published:** 2022-07-28

**Authors:** Federico Soria, Salvador David Aznar-Cervantes, Julia E. de la Cruz, Alberto Budia, Javier Aranda, Juan Pablo Caballero, Álvaro Serrano, Francisco Miguel Sánchez Margallo

**Affiliations:** 1Jesus Uson Minimally Invasive Surgery Centre Foundation, Endoscopy-Endourology Department, 10071 Cáceres, Spain; jecruz@ccmijesususon.com (J.E.d.l.C.); javierarandap@gmail.com (J.A.); msanchez@ccmijesususon.com (F.M.S.M.); 2Biotechnology Department, IMIDA, 30150 Murcia, Spain; sdac1@um.es; 3Urology Department, Polytechnic and University La Fe Hospital, 46026 Valencia, Spain; alberto.budia@hotmail.com; 4Urology Department, University General Hospital, 03010 Alicante, Spain; juanpablocaballero@gmail.com; 5Urology Department, University Clinico San Carlos, 28040 Madrid, Spain; alvaro.serrano.p@gmail.com

**Keywords:** ureteral stent, biodegradable stent, drug eluting stent, chemotherapy, UTUC

## Abstract

A major limitation of the treatment of low-grade upper tract urothelial carcinoma is the difficulty of intracavitary instillation of adjuvant therapy. Therefore, the aim of this in vitro study was to develop and to assess a new design of biodegradable ureteral stent coated with a silk fibroin matrix for the controlled release of mitomycin C as a chemotherapeutic drug. For this purpose, we assessed the coating of a biodegradable ureteral stent, BraidStent^®^, with silk fibroin and subsequently loaded the polymeric matrix with two formulations of mitomycin to evaluate its degradation rate, the concentration of mitomycin released, and changes in the pH and the weight of the stent. Our results confirm that the silk fibroin matrix is able to coat the biodegradable stent and release mitomycin for between 6 and 12 h in the urinary environment. There was a significant delay in the degradation rate of silk fibroin and mitomycin-coated stents compared to bare biodegradable stents, from 6–7 weeks to 13–14 weeks. The present study has shown the feasibility of using mitomycin C-loaded silk fibroin for the coating of biodegradable urinary stents. The addition of mitomycin C to the coating of silk fibroin biodegradable stents could be an attractive approach for intracavitary instillation in the upper urinary tract.

## 1. Introduction

Urothelial carcinomas are the sixth most common tumors in developed countries. They may be located in the lower urinary tract, bladder or urethra, as well as in the upper urinary tract, ureter, and pyelocaliceal system. Bladder tumors are responsible for 90–95% of urothelial carcinomas and are the most common neoplasm of the urinary tract. Upper tract urothelial carcinomas (UTUC) are uncommon and represent only 5–10% of total cases, with a yearly incidence of almost 2 cases per 100,000 population [1].

Patients with UTUC may have one of two types of urothelial carcinoma: low-grade or high-grade. It is very useful to stratify the risk of UTUC into low- and high-grade in order to identify those patients who will benefit from nephron-sparing surgery and those who should instead undergo radical treatment, such as nephroureterectomy [2].

In order to categorize a tumor as low risk, each of the following characteristics must be met: unifocal disease, tumor size <2 cm, negative urine cytology for high-grade tumor, positive biopsy for low-grade tumor, and non-invasive appearance on a computed tomography (CT) scan [3]. These patients may benefit from minimally invasive treatment using a transurethral endoscopic approach with a semi-rigid or flexible endoscope. Additionally, an adjuvant instillation of mitomycin C (MMC) has provided promising results in previous studies and may reduce the risk of urothelial recurrence and progression in patients affected by low grade UTUC [3,4]. Unfortunately, there is currently no suitable procedure for adjuvant chemotherapeutic intracavitary instillation after holmium laser fulguration of UTUC tumor lesions [5]. This is due to the difficulty of intracavitary chemotherapy instillation in the upper urinary tract, given the washout effect of urine production at the renal level and the low storage capacity of the upper urinary tract [6]. As a result, many patients do not benefit from an adjuvant chemotherapy instillation procedure, which leads to worse results in this group of patients.

A recent meta-analysis highlighted that novel drug delivery technologies promise to change this paradigm by favoring drug exposure in the upper tract, leading to higher treatment efficacy [7]. One of the new technologies for chemotherapy instillation in the upper urinary tract is a gel loaded with 4 mg/mL MMC. When it is instilled, it is liquid, but at body temperature it gels and is removed within 4–6 h, through the urinary tract. It is used for primary chemo-ablation. A single-arm, open-label, phase III clinical trial has shown very encouraging results, which have been evaluated up to 12 months. This system was approved by the Food and Drug Administration (FDA, Silver Spring, MD, USA) 2 years ago; it shows a successful response in 59% of patients but 27% experience severe side effects, and the administration schedule is very complicated for patients [8].

A further new development in the delivery of cytostatics to the upper urinary tract comes from vascular stents, called ‘drug eluting stents’ [9]. Our research group has developed a biodegradable ureteral stent known as the BraidStent^®^ [10]. This platform has been coated with a multi-layered silk fibroin (SF) protein coating for the controlled release of MMC. SF has previously demonstrated outstanding properties as a drug eluting coating, such as for stent applications, based on its useful mechanical properties and biological outcomes [11,12,13]. The scientific literature has high expectations for fibroin, reporting that it represents a new biomaterial with improved mechanical properties as a scaffold. It is expected to overcome the limitations of current biomaterials for stent coating, both as a polymer carrier and because of its demonstrated biocompatibility and easy processability. In fact, many studies have shown that SF is more biocompatible than other currently-used polymeric degradable biomaterials, such as PLGA (poly-lactic-co-glycolic acid), PLA (polylactic acid), and PGA (polyglycolic acid) [11,14,15].

The aim of this comparative in vitro experimental study was to develop and to assess a new design of biodegradable ureteral stent coated with a SF matrix for the controlled release of MMC for intracavitary instillation in the adjuvant treatment of UTUC.

## 2. Materials and Methods

The experimental study was organized into three protocols.

**Experimental protocol I**. In the first protocol, the aim was to compare two combinations of biodegradable polymers and copolymers for the manufacture of the biodegradable ureteral stent before coating it with SF.

**Materials for stent preparation.** Three polymers and copolymers were selected for this purpose: Glycomer^TM^ 631 (Biosyn suture by Covidien, Minneapolis, MN, USA), PGA (Safil^®^ Quick suture by B. Braun, Secaucus, NJ, USA), and poly-4-hydroxybutyrate (Monomax^®^ suture by B. Braun Surgical, Barcelona, Spain). All three biomaterials are derived from biocompatible and biodegradable surgical sutures. The search for the right combination of polymers for the biodegradable ureteral stent aimed to produce a stent that degrades within 7–8 weeks in a progressive manner to avoid obstructive degradation in future in vivo studies.

In order to compare the degradation rate, 3-cm long fragments of biodegradable ureteral stents were developed using the following combinations: Group BraidStent-1, a long-term braided stent with Glycomer^TM^ 631 and poly-4-hydroxybutyrate; and group BraidStent-2, a short-term braided stent with Glycomer^TM^ 631 and PGA. The ratio in the composition of the polymers of each stent was always kept constant in their manufacture: Glycomer^TM^ 631 (54%); PGA and poly-4-Hydroxybutyrate (46%).

Five samples of each of the types of stent fragments were developed, giving a total of 10 samples. These were placed in watertight tubes with artificial urine (human synthetic urine, BioIVT, Royston, UK), pH: 6.7 specific gravity 1.008 (FDA registered) for screening studies for high performance liquid chromatography with diode-array detection (HPLC-DAD). To investigate the degradation rate, the stent fragments were dipped in 5 mL of artificial urine (AU) and incubated in an orbital shaker-incubator under mimicked biological conditions (36.5 °C with 5% CO_2_, at 90 rpm) until complete stent biodegradation, with daily AU changes. Four follow-ups were performed in each group: T0—start of the study; T1—day of onset of macroscopic degradation; T2—day on which we macroscopically detected that the stent had degraded by 50%; and T3—complete stent degradation. The changes in pH, manifestation of nitrites, weight of the wet stent, and days at which the described changes appeared were assessed. The average thickness of each stent was also determined at baseline (T0) (Figure 1).

**Experimental protocol II.** Once the study corresponding to protocol I had been completed and the most suitable polymer/copolymer combination for the development of the BraidStent^®^ was known, we proceeded to the next step, which consisted of the SF coating of the BraidStent^®^ (the BraidStent-SF group).

**Materials for stent preparation and SF coating.** Cocoons of *Bombyx mori* were obtained from worms reared in the sericulture facilities of the IMIDA, Biotechnology Department (Murcia, Spain). Cocoons were chopped up and boiled in 0.02 M Na_2_CO_3_ for 30 min in order to eliminate the sericin. Then, the raw SF was rinsed with distilled water and dried at room temperature for 3 days. Subsequently, SF was dissolved in 9.3 M LiBr (Acros Organics) for 3 h at 60 °C, yielding a 20% *w*/*v* dissolution that was dialyzed against distilled water for 3 days (Snakeskin Dialysis Tubing 3.5 KDa MWCO, Thermo Scientific, Waltham, MA, USA) with 8 total water changes (at 4 °C) [16]. The resultant 7–8% *w*/*v* SF solution was recovered and used for the preparation of the coated BraidStent-SF stents as explained below, adjusting the concentration to 7% *w*/*v* before use. The protocol was adapted from the methodology proposed by Rockwood et al. for the manufacture of fibroin tubes by means of a dipping technique using alternate baths of aqueous fibroin and methanol [17].

The BraidStent-SF group was composed of the same number of stent fragments (five) and followed the same experimental protocol, with the same follow-ups and assessment of the same variables as in protocol I (Figure 2).

**Characterization**. In order to assess the appropriate SF coating on the stent surface, a comparative study was carried out with three samples per group, where BraidStent-2 was compared to BraidStent-SF with sulforhodamine B (SRB) (a fluorescent aqueous marker) coating added to visualize the homogeneity of the SF coating by fluorescence microscopy.

**Experimental protocol III**. Following the evaluation of the BraidStent-SF, it was loaded with two different MMC formulations (BraidStent-SF-MMC1 and BraidStent-SF-MMC2) to assess the release concentration and MMC release time. The BraidStent-SF-MMC1 and 2 groups were composed of the same number of stent fragments, 5 per group, and followed the same experimental protocol and follow-ups as in protocols I and II.

**Materials for stent preparation and SF and MMC coating.** For the BraidStent-SF-MMC1 group, the coating of the stents used powdered 10 mg MMC to produce the intravenous injectable solution commonly used in the medical field (INIBSA, Barcelona, Spain); the vials contained sodium chloride as an excipient (9.5 mg of sodium per 1 mg of MMC). For this, a 7% *w*/*v* fibroin solution containing 5 mg/mL of MMC was prepared by dissolving the drug for 30 min under orbital agitation at 120 rpm. At the same time, the same protocol was performed using absolute methanol, dissolving MMC at the same concentration and for the same period of time. These two starting solutions were used to alternately bathe the stent fragments, by dipping for 5 s in the first solution (of aqueous fibroin) and then in the second (containing methanol), for the same time, and letting them dry slightly for 1 min before repeating the procedure. These coating cycles were carried out in this BraidStent-SF-MMC1 group 10 times. The containers used for this purpose were 5 mL glass tubes (Figure 3).

For the BraidStent-SF-MMC2 group, the concentration employed for both the fibroin and methanol solutions was 10 mg/mL, and 10 dip coating cycles were performed. In this group, 70 mg of pure MMC (Mitomycin CRS, Sigma-Aldrich, Steinheim am Albuch, Germany) was used, without any excipient (Figure 3).

The stents were completely stable in room air. However, the MMC coating needed special care, i.e., it must be packaged in a lightproof for preservation and transportation to ensure its physical and chemical stability. MMC shows some light sensitivity, so reasonable steps to minimize light exposure should be taken.

**MMC assessment.** MMC concentrations were assessed every six hours until there was no analytical evidence of its detection. Urine from each follow-up was replaced and analyzed. MMC was released from the SF matrix due to solubility events of the SF matrix in the urinary environment. The permeability and release kinetics of the MMC depends on the SF coating and relates to the percentage of the beta-sheet structure. Increasing the crystallinity (methanol immersion) of the SF beta sheet decreases the release rate and increases the release duration.

The HPLC-DAD method was an isocratic method with a mobile phase of acetonitrile: ultrapure water 80:20 (*v*/*v*) at a flow rate of 1 mL/min, and separation was performed on a LUNA C18 250 mm, 4.6 mm, 5 µm column at 30 °C temperature. A total of 10 µL of sample was injected. The MMC was detected with a diode array detector (DAD) at 365 nm (1260 Infinity II Prime LC System, Agilent Technologies, Santa Clara, CA, USA).

**Statistical analysis.** Statistical analysis was performed with the SPSS 25.0 program for Windows (IBM, Armonk, NY, USA). The variables studied were pH, weight of the stents in g, degradation rate expressed in days, stent width in mm, and artificial urine concentration of MMC in mg/L. Variables are shown as their mean ± standard deviation. The normality of the data was analyzed using the Shapiro–Wilk test. For data that followed a normal distribution, the intra-group and intra-phase distributions were analyzed using a one-factor ANOVA. Post-hoc analysis was performed using Tukey’s HSD (honestly significant difference) test.

For variables in which the data were not normally distributed, the comparison between groups in each phase was carried out using the Kruskal–Wallis test. In the event of statistical significance, the corresponding post-hoc pairwise comparison was carried out. The trends of the variables throughout the follow-ups concerning the effect of the factors time and group, were analyzed by means of a General Linear Model (GLM) Repeated Measures. Again, the post-hoc analysis was performed using Tukey’s HSD. In certain cases, to evaluate the trend of each group along time, a Friedman test or a Wilcoxon test was performed, depending on the number of follow-ups included in the analysis. In addition, the concentration of Mitomycin C released by the two groups, BraidStent-SF-MMC1 and BraidStent-SF-MMC2, was analyzed either via a t-test for independent samples or a Mann–Whitney test, depending on the normality of data. Confidence intervals were set at 95% and significance was determined by *p*-values less than 0.05.

## 3. Results

**Protocol I.** We found significant differences between the two groups for the variable ‘stent weight’ at T0 and T1. The combined weight of polymers and copolymers in BraidStent-1 was significantly higher than in BraidStent-2. BraidStent-1 showed a weight loss of 11.11% and BraidStent-2 showed a weight loss of 14.81% between the baseline study and the macroscopic onset of degradation (Figure 4). The duration of degradation variable shows a statistically significant difference between both groups at T1 and T3, but not at T2 (Figure 5). BraidStent-2 started degrading later, but between T2 and T3, there was an acceleration of the hydrolysis process, resulting in a shorter time to complete stent degradation. BraidStent-2 fit the criterion of degradation in the first 7–8 weeks (Figure 5). The difference in the pH of the medium in protocol I showed statistical significance at T1 and T2. The stent degradation metabolites of BraidStent-1 and BraidStent-2 caused overt acidification of the medium, which was more marked with BraidStent-1. At T2 and T3, the pH returned nearly to basal levels. Using Friedman’s statistical test, we assessed the trend over the different phases and whether the changes in pH within each group were uniform (Figure 6). In both groups, the pH trend was not uniform and showed significance throughout the different phases (BraidStent-1 group *p* = 0.006 and BraidStent-2 group *p* = 0.009). None of the samples were positive for nitrites on urinalysis.

**Protocol II.** For this protocol, the BraidStent-2 group was selected as it met the inclusion criteria of complete degradation before 8 weeks. The BraidStent-SF group used the same combination of polymers and copolymers as in the BraidStent-2 group, dip-coated with SF. The characterization results of the comparative study between BraidStent-2 and BraidStent-SF SRB coated show, after evaluation with fluorescence microscopy, that the SF coating of the stent is uniform (Figure 7).

The BraidStent-SF group did not show statistically significant changes between T0 and T1, with a decrease in the weight of 11.36% between the two phases (Figure 4 and Figure 8). With regard to degradation time, the addition of the SF coating led to significant differences compared to the BraidStent groups, significantly increasing the degradation time compared to the stent without the SF coating at T1-T2-T3 (Kruskal–Wallis H test, *p* = 0.002). The Friedman test determined that the trend in pH between the study phases showed statistical significance (*p* = 0.016). As in groups BraidStent-1 and -2, there was acidification of the urinary medium at the beginning of degradation, which subsequently recovered to basal levels at T3 (Figure 6). None of the samples were positive for nitrites on urinalysis, ruling out bacterial contamination.

**Protocol III.** Following the addition of the two MMC formulations to the stent, the weight showed statistically significant changes between the BraidStent-SF-MMC1 and the BraidStent-SF-MMC2 groups, with a greater weight in the latter group at T0 and T1. Both groups showed statistically significant differences compared to the SF-coated stent without MMC; thus, the addition of MMC significantly increased the weight of the stent (Figure 4). Moreover, stent thickness showed statistically significant differences between the BraidStent-SF group and the two groups coated with MMC (Figure 9). There was no statistically significant difference between the BraidStent-SF-MMC1 and the BraidStent-SF-MMC2 groups with regard to the manifestation of the different degradation phases, however, when both groups were compared with BraidStent-SF, statistical significance was found at T2, with this latter group showing a much faster degradation at this follow-up (Figure 5). The BraidStent-SF-MMC1 group, in contrast to BraidStent-SF-MMC2, shows a uniform trend in its variations throughout the urinary pH study (Friedman’s test, *p* = 0.357) (Figure 6). Both stent groups underwent the same number of coating cycles, 10, although the initial concentration of MMC in BraidStent-SF-MMC1 was 10 mg compared to 70 mg in BraidStent-SF-MMC2. In addition, the joint dilution of SF and MMC was 5 mg/mL and 10 mg/mL, respectively. The kinetics of MMC in both groups was fully released within the first 12 h, with no analytical evidence of MMC afterward. We found statistically significant differences in the concentration of MMC released at 12 h between the groups, with a higher urine concentration of MMC in the BraidStent-SF-MMC2 group (Table 1). Statistically significant differences were also found within each group between 6 and 12 h. None of the samples were positive for nitrites on urinalysis.

## 4. Discussion

In response to the clear need to design new intracavitary chemotherapy instillation systems for adjuvant treatment of low-grade UTUC, gels and ureteral stents that can deliver chemotherapeutics have been developed [7,8,9,18,19]. MMC-eluting gels are already FDA-approved; these represent an important advance, despite having demonstrated a high complication rate in the first clinical study [8,18]. There is also an in vitro study on a chemotherapy-eluting ureteral stent reported in the scientific literature [9,19]. These two developments for intracavitary instillation of chemotherapy present important limitations at the clinical level, since, in the case of the gel, it is necessary to introduce a ureteral catheter for instillation, and in the case of the ureteral stent, it is necessary to remove it cystoscopically. The design we assessed in this experimental study uses a biodegradable ureteral stent able to release MMC as a chemotherapeutic agent. This avoids the necessary removal of plastic ureteral stents, which would reduce health care costs and improve the quality of life of patients.

The optimal combination of polymers and copolymers in the stent is determined by the requirements of their placement in the upper urinary tract [10,20]. The main current limitations to the development of biodegradable ureteral stents (BUS) are related to the control of the degradation rate and the size of the degradation fragments [10,20,21,22,23]. For this reason, a combination of polymers and copolymers with different degradation rates (always with degradation mainly by hydrolysis when placed in urine) are chosen to produce BUS. PGA is a synthetic, biodegradable, rapidly-degrading polymer, with glycolic acid as a metabolite. Poly-4-hydroxybutyrate is a long-term absorbable biomaterial, in which degradation occurs by hydrolysis and enzymatic pathways. 4-hydroxybutyric acid is a natural metabolite, which is primarily converted to carbon dioxide and water [24]. Finally, Glycomer^TM^ 631 shows a degradation rate intermediate between these two options [25]. The application of polymers used in surgical sutures has been previously described. Due to their biocompatibility and ease of biodegradation, they may be a suitable material for drug delivery by techniques that include electrospinning, melt-extrusion, and coating [26].

For this reason, the two stent groups (BraidStent 1 and 2) use stents composed of a combination of two synthetic polymers with different degradation rates. This allows the degradation to be controlled as their biodegradation starts at different times, thus allowing the stent to fulfil its function as a scaffold while gradually reducing its mass, producing smaller fragments than if the stents were composed of a single degradable biomaterial. In protocol I, we found that both groups showed significant differences at T0 and T1 with respect to stent weight, but not stent thickness (Figure 4 and Figure 9). We believe that the degradation characteristics of poly-4-hydroxybutyrate are the cause of the significantly slower biodegradation, although it should be noted that the degradation fragments were similar in both groups. An in vitro study by Barros et al. used artificial urine to assess a BUS from natural origin polymers (biopolymers) such as alginate as well as gellen and their blends with gelatine; it was found that the degradation duration ranged from 14 up to 60 days. This is in agreement with our results with synthetic polymers from protocol I, i.e., 52–77 days. Unlike synthetic polymers, biopolymers present better elasticity, biocompatibility, interface lubricity, and resistance to biofilm formation and encrustation [27].

One factor to consider in the assessment of new biodegradable stents in the urinary tract is that most synthetic polymers show a higher degradation rate at acidic pH than at physiological pH, which affects the ability to adjust the degradation time of the stent to the demands of patients [24,28]. Our experimental study demonstrates that, in all tested groups, excluding the BraidStent-SF-MMC1 group, the pH decreased significantly between baseline and T1, with a greater decrease in pH in the non-SF groups (pH < 6.0) (Figure 6). However, the range of pH changes found in our study, irrespective of the groups, remains within normal values for human beings. This is very relevant as significant alterations in urinary pH are associated with stent encrustation or crystal deposition that can lead to urinary lithiasis.

SF represents a new biomaterial, which, according to the scientific literature, can solve weaknesses related to stent coating, as it has the ability to carry drugs, control drug release kinetics, and improve the internal scaffold function [11]. The biocompatibility and biosafety of SF have been demonstrated previously, and the FDA has approved this polymer matrix for use in medical devices [12]. SF stent coating has been used for the delivery of various drugs, such as heparin, paclitaxel, clopidogrel, curcumin, 5-fluorouracil, sirolimus, emodin, theophylline, amoxicillin, and salicylic acid [11,12,13,29,30,31]. In the current experimental study, we added MMC to this series of drugs, which had not previously been evaluated as a drug to be included in an SF coating. Our results confirm that SF is useful for carrying and releasing MMC over a time period that could correspond to the postoperative application of chemotherapy (Table 1). Additionally, its biodegradation capacity allows the ureteral stent designed in our study to completely degrade. This degradation of the stent and its coating took place in a urinary environment, and it was achieved due to the layer-by-layer distribution with which it was designed. It is very important to control the changes in SF crystallization because the crystalline domains of SF are mainly formed by its beta-sheet structure, which contributes to MMC release. Our coating technique, by dipping the aqueous SF in absolute methanol, allowed the insolubilization process to increase the beta-sheet content of the regenerated SF. In our study, this allowed for the deposition of SF layer by layer, ensuring the encapsulation of MMC and its gradual and programmed release in the first 12 h, in order to achieve a suitable protocol for patients. In our methodology, to avoid losses in the encapsulation of MMC in the SF layers during methanol dipping, MMC was incorporated not only in the SF solution, but also in the methanol solution, at the same concentration. This represents an innovation with respect to the SF dip-coating techniques reported in the scientific literature.

An interesting attribute provided by SF is its ease of processability in different formats such as gels, membranes, coatings, and scaffolds [32]. This property was used in our study for the BraidStent-2 coating. Despite the statistically significant increase in thickness that SF and MMC caused, it is important to highlight that though thickness is below the size used in patients so it will not be an obstacle to clinical application [33].

Comparisons of this new design (BraidStent-SF-MMC1 or -2) with other biodegradable ureteral stents made of PGA and Glycomer^TM^ 631 have been experimentally evaluated in the literature [21,22,23]. We found that, as with Braidstent groups 1 and 2 in the current study, coating with SF and MMC caused a significant reduction in the rate of ureteral stent degradation, from 6–7 weeks to 13–14 weeks (Figure 5) [21,22,23]. Despite significant differences in baseline thickness and weight between the BraidStent-SF, and the BraidStent-SF-MMC1- and -2 groups, no differences in overall degradation time were seen (Figure 4, Figure 5, and Figure 9). This may be due to the fact that, in both MMC formulations, complete release occurred within the first 12 h, a fact verified by HPLC-DAD. On the other hand, in a study assessing a very promising compound, such as Titanium dioxide nanoparticles to inhibit tumor development, which has shown in cell culture and in animal models similar results to current chemotherapy (Doxorubicin) but which reduces the side effects associated with conventional therapies. The researchers find, as in our study, that exposure at 6 and 12 h reach their maximal uptake and relate this to anti-tumor efficacy [34]. Regarding this MMC release rate, we found a huge difference between the two formulations, especially at 12 h, with a significantly higher concentration of MMC released by the BraidStent-SF-MMC2 group (Table 1). In this regard, unfortunately our study did not allow us to clarify whether the difference in the concentration of MMC released between the two groups depended exclusively on the different concentration of MMC added to the SF, or whether it was due to the use of pure MMC in the BraidStent-SF-MMC2 group. However, we did find greater difficulties in the manufacture of stents when MMC contained sodium chloride as an excipient compared to pure MMC, which did not cause precipitation of the dip solution. This is an aspect that will be assessed in future studies.

## 5. Conclusions

The present study demonstrated the feasibility of using MMC-loaded SF for the coating of biodegradable urinary stents. Coating with a fibroin matrix allowed for the controlled and programmed release of MMC but delayed the complete biodegradation of the ureteral stent. The addition of MMC to the coating of SF stents could be an attractive approach for intracavitary instillation in the upper urinary tract.

## Figures and Tables

**Figure 1 polymers-14-03059-f001:**
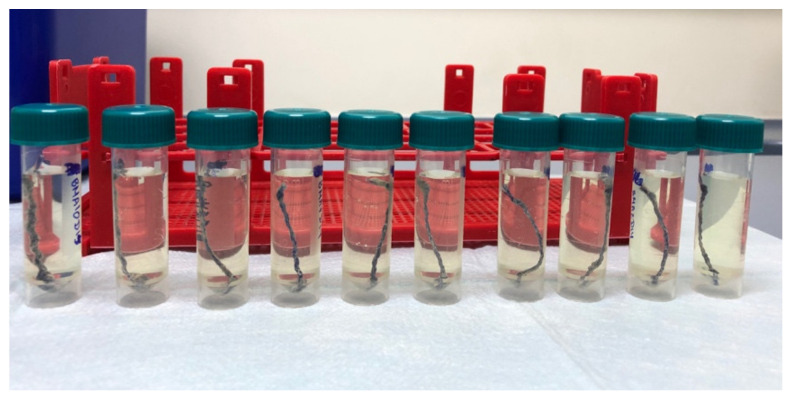
Ureteral stents fragments dipping in artificial urine (Day 6).

**Figure 2 polymers-14-03059-f002:**
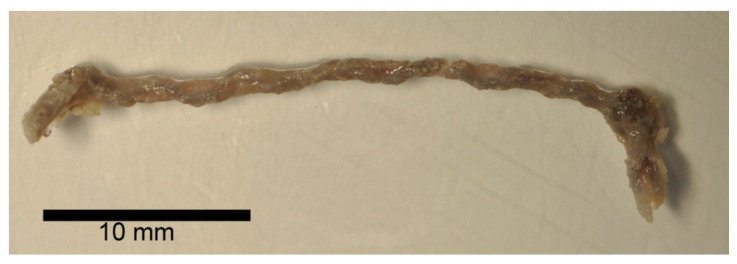
BraidStent-SF sample. The SF coating of the stent can be appreciated.

**Figure 3 polymers-14-03059-f003:**
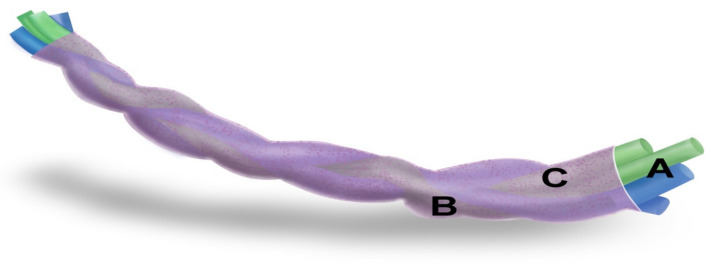
Illustration of BraidStent-SF-MMC. **A**—BraidStent; **B**—SF coating; **C**—MMC embedded in SF matrix.

**Figure 4 polymers-14-03059-f004:**
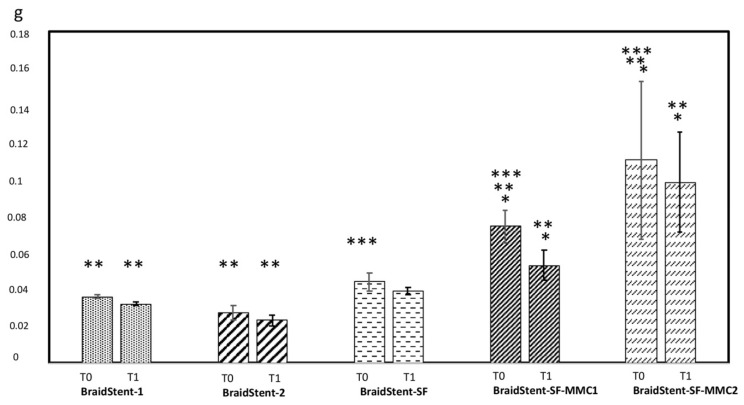
Assessment of weight loss (grams) between the start of the study (T0) and the start of stent degradation (T1). Values indicate mean ± SD. Significance of the values between each tested groups at T0 and T1 was determined by the Kruskal–Wallis H test and post hoc pairwise comparison. Wilcoxon test was used to analyze weight loss from T0 to T1 within each group. Significant differences are expressed as: intra-groups (* *p* < 0.05); inter-groups (** *p* < 0.05) (BraidStent-1 vs. BraidStent-2; BraidStent-SF-MMC1 vs. BraidStent-SF-MMC2); inter-groups (*** *p* < 0.05) (BraidStent-SF vs. BraidStent-SF-MMC1 and BraidStent-SF-MMC2).

**Figure 5 polymers-14-03059-f005:**
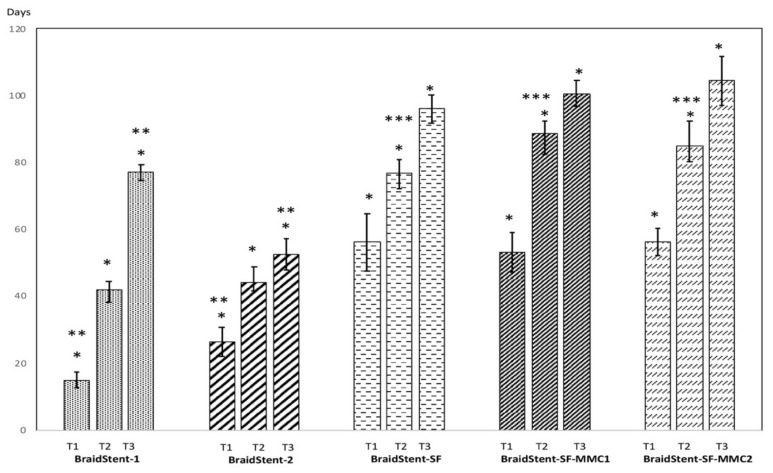
Degradation rate assessment (days) between T1 (onset of degradation); T2 (50% of stent degradation); T3 (complete stent degradation). Values indicate mean ± SD. Significance of the values among tested groups within each follow-up was determined by the Kruskal–Wallis H test and post hoc pairwise comparison. The effect of group and time along follow-ups from T1 to T3 was assessed via a GLM Repeated Measures and pairwise comparison via Tukey’s test. Statistical significance is depicted as follows: intra-groups (* *p* < 0.05); inter-groups (** *p* < 0.05) (BraidStent-1 vs. BraidStent-2; BraidStent-SF-MMC1 vs. BraidStent-SF-MMC2); inter-groups (*** *p* < 0.05) (BraidStent-SF vs. BraidStent-SF-MMC1 and BraidStent-SF-MMC2).

**Figure 6 polymers-14-03059-f006:**
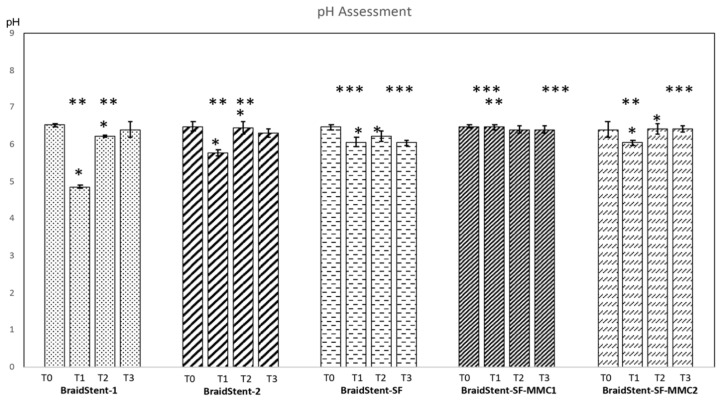
Assessment of pH throughout the study until complete stent degradation. T0 (start of study); T1 (onset of degradation); T2 (50% of degradation); T3 (complete stent degradation). Values indicate mean ± SD. Significance of the values between each tested groups within each phase of the study was determined by the Kruskal–Wallis H test and post hoc pairwise comparison; while a General Linear Model (GLM) Repeated Measures was used to evaluate the effect of both group and time (from T0 to T3) in the trend of pH, with the corresponding post-hoc analysis by Tukey test. Statistical significance is indicated in the figure as follows: intra-groups (* *p* < 0.05); inter-groups (** *p* < 0.05) (BraidStent-1 vs. BraidStent-2; BraidStent-SF-MMC1 vs. BraidStent-SF-MMC2); inter-groups (*** *p* < 0.05) (BraidStent-SF vs. BraidStent-SF-MMC1 and BraidStent-SF-MMC2). Friedman test shows that only the BraidStent-SF-MMC1 group is homogeneous along follow-ups.

**Figure 7 polymers-14-03059-f007:**
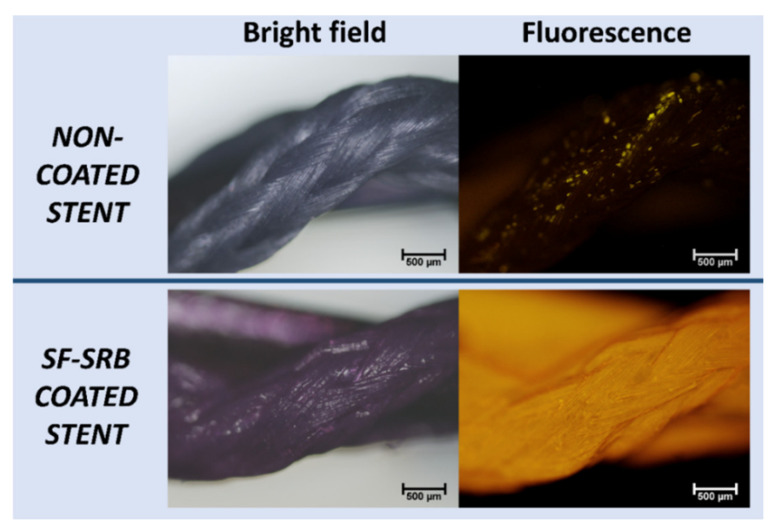
Assessment of SF coating with SRB fluorescent dye. Comparative study between BraidStent-2 (non-coated stent) and BraidStent-SF with SRB coating. Fluorescence microscopy showed adequate homogeneity of SF coating.

**Figure 8 polymers-14-03059-f008:**
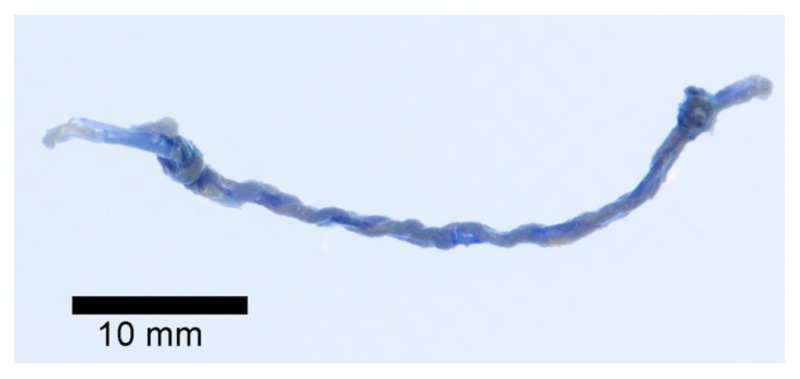
BraidStent-SF sample in T2. It can be appreciated that the SF coating has practically disappeared.

**Figure 9 polymers-14-03059-f009:**
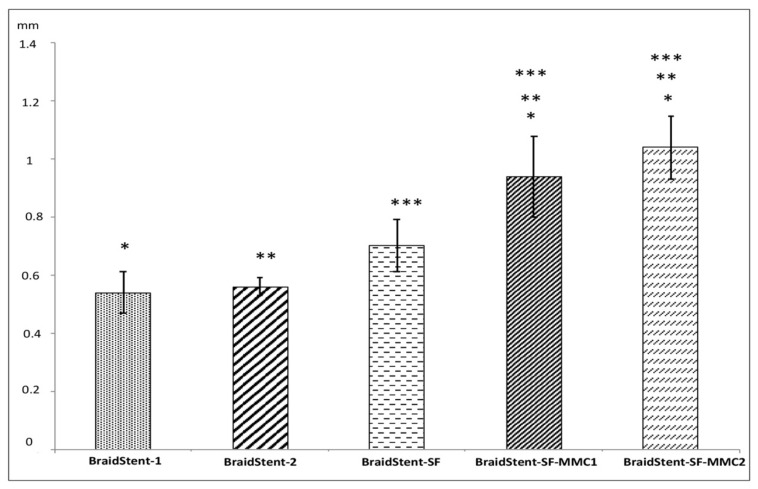
Stent thickness in the experimental groups at baseline (T0) (mm). Values indicate mean ± SD. Significance of the values between each tested group was determined by a one-way ANOVA and a post hoc Tukey test: (* *p* < 0.05) (BraidStent-1 versus the rest of experimental groups); (** *p* < 0.05) (BraidStent-2 versus the rest of experimental groups); (*** *p* < 0.05) (BraidStent-SF versus the rest of experimental groups).

**Table 1 polymers-14-03059-t001:** Mitomycin C concentration release rate (mg/L). MMC release is assessed every 6 h in artificial urine in BraidStent-SF-MMC1 and BraidStent-SF-MMC2 stents, until it is not detected by HPLC-DAD. The variable mg/l MMC at 6 h follows a normal distribution but not at 12 h. A Student’s *t*-test for independent samples was performed in order to determine the differences at 6 h between groups. As for the study at 12 h, a Mann–Whitney test was performed. The trend in MMC concentration over the 6 and 12 h, between groups and within each group, was analyzed using a GLM Repeated measures. Superscripts refer to statistical significance between the values, namely inter- or intra-group. a-b-c (*p* < 0.01).

mg/L	6 h	12 h	18 h	24 h
BraidStent-SF-MMC1	10.98 ± 3.73 ^a^	2.33 ± 3.05 ^ac^	0	0
BraidStent-SF-MMC2	7.67 ± 1.39 ^b^	56.08 ± 4.76 ^bc^	0	0

## Data Availability

Not applicable.

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
