# Peer review of "Assessment of a Coated Mitomycin-Releasing Biodegradable Ureteral Stent as an Adjuvant Therapy in Upper Urothelial Carcinoma: A Comparative In Vitro Study"

_polymers, 2022, doi:10.3390/polym14153059_

Round 1

Reviewer 1 Report

Dear authors,

Your work and toic are interesting. It fits the scope of Polymers. You report in vitro that (i) silk fibroin matrix is able to coat the biodegradable stent and release mitomycin for between 6 and 12 h in the urinary environment; (ii) there was a significant delay in the degradation rate of silk fibroin and mitomycin-coated stents compared to bare biodegradable stents, from 6–7 weeks to 13–14 weeks; (iii) the feasibility of using MMC-loaded silk fibroin for the coating of biodegradable urinary stents and an attractive approach for intracavitary instillation in the UTUC...

First of all, there are many points to consider prior to consider the article for publication in Polymers:

1- Typos and grammatical errors, spelling, syntax, and spacing: Please go through the entire manuscript to revise it carefully

e.g. L 45... shall be  Tomography; L. 88 remove the space betweern PGA and (Safil...)

2- Introduction: add the pertinence of using MMC in UTUC. Also, you may add possible works using algal-based polymers for developping biodegradable stents  and compare the materials used in your study with such marine polymers;

3 - Methodology: lack of physical characterizations to confirm the coating of MMC as well as the final product developed.

In this regards for the methological aspects), I suggest to refer and cite this works:

Razzaq A, Khan ZU, Saeed A, Shah KA, Khan NU, Menaa B, Iqbal H, Menaa F. Development of Cephradine-Loaded Gelatin/Polyvinyl Alcohol Electrospun Nanofibers for Effective Diabetic Wound Healing: In-Vitro and In-Vivo Assessments. Pharmaceutics. 2021 Mar 7;13(3):349. doi: 10.3390/pharmaceutics13030349. PMID: 33799983; PMCID: PMC7998169.

4- Results: the figure captions are incomplete; statistics shall be informative...

best,

the reviewer

Author Response

R1.1. typos and gramatical errors, spelling, syntax and spacing: Please go through the entire manuscript to revise it carefully.

We thank the reviewer for pointing this out. We have revised the manuscript and resubmitted it to a proof-reading service to improve its comprehensibility (Proof-reading-service.com LTD, Devonshire).

R.1.2. Add the pertinence of using MMC in UTUC.

Additional text has been added to the manuscript to justify the use of MMC in these patients:

“These patients may benefit from minimally invasive treatment using a transurethral endoscopic approach with a semi-rigid or flexible endoscope. Additionally, an adjuvant instillation of mitomycin C (MMC) has provided promising results in previous studies and may reduce the risk of urothelial recurrence and progression in patients affected by low grade UTUC [3-4]. Unfortunately, there is currently no suitable procedure for adjuvant chemotherapeutic intracavitary instillation after holmium laser fulguration of UTUC tumour lesions [5].” Line 43-47.

-Add works using algal-based polymers for developing BUS.

A comparison of Barros AA's research with biopolymers and our results in the protocol-I has been added in the Discussion section.

“An in vitro study by Barros AA et al. used artificial urine to assess a BUS from natural origin polymers (biopolymers) such as alginate, gellen and their blends with gelatine; it was found that the degradation duration ranged from 14 up to 60 days. This is in agreement with our results with synthetic polymers from protocol-I, i.e. 52-77 days. Unlike synthetic polymers, biopolymers present better elasticity, biocompatibility, interface lubricity and resistant to biofilm formation and encrustation [27].”

R1.3. Methodology: lack of physical characterization to confirm the coating of MMC as well as the final product developed.

We thank the reviewer for his/her very useful feedback. You are absolutely right, we have not performed electron microscopy studies to evaluate the surface of the coating, but we have performed other assessments to confirm adequate BraidStent® coating. All the stents in the different groups were weighed and measured for thickness before starting the study, so that we can assess the increases in weight and thickness corresponding to each type of coating (the SF and the SF with the MMC) (Table 1 and 2). On the other hand, in the initial study (data not included), after coating BraidStent-2 with SF, we added sulforhadamine B (SRB) in order to visualise the homogeneity of SF coating by fluorescence microscopy. The assessment of coated stents under the microscope shows the homogeneous layer of SF containing SRB. The concentration of Mitomycin C (HPLC-DAD) released was evaluated in the urine of each of the refills (6h, 12h, 18h, 24h).

R.1.4. Results: the figure captation are incomplete; statistics shall be informative…

Some figures have been changed to be more illustrative and figure 4 has been removed. The captions have been revised according to the reviewer's criteria.

PLEASE SEE THE ATTACHMENT

Reviewer 2 Report

The manuscript by Soria et al. described the fabrication of drug loaded ureteral stent for upper urothelial carcinoma. The authors performed in vitro degradation tests using weight changes of the stents. Overall, the manuscript fits the scope of the Journal. However, there are some components missing in the study. It is encouraged that the authors considered the changes to improve the quality of the manuscript.

(1)  In Figure 1, some portions of the ureteral stents were not fully immersed in the artificial urine. How does this affect the in vitro degradation rate?

(2)  For experimental protocol-I, please indicate the compositions of Group BraidStent-1 and Group BraidStent-2. For example, how much of GlycomerTM 631 is combined with Poly-4-Hydroxybutyrate and how much of GlycomerTM 631 is combined with PGA?

(3)  Please provide weight change data from protocol-I to protocol-II, BraidStent-2 group and BraidStent-SF group.

(4)  The authors mentioned in Section 2 on the time points of the degradation as follows: “Four follow-ups were performed in each group: T0 – start of the study; T1–start of degradation under macroscopic control; T2–50% stent degradation; and T3–complete stent degradation.” This statement is not consistent with Figure 5. Please reconsider the expression of time points. Also, in Figure 5, what is the title for the y-axis?

(5)  For protocol-III, what are the drug loadings for the BSFM1 and BSFM1 groups?

(6)  How was the in vitro MMC release was conducted and how were the MMC concentrations at various time points analyzed?

Author Response

R.2.1. In Figure 1, some portions of the ureteral stents were not fully immersed in the artificial urine. How does this affect the in vitro degradation rate?

Thank you very much for pointing this out.

The figure has been changed to avoid misleading readers. The tubes are in vertical position only for taking the pictures, as in the orbital shaker-incubator they are in horizontal position at 90 rpm stirring, which ensures that the stent is completely dipped in the artificial urine.

R.2.2. For experimental protocol-I, please indicate the compositions of Group BraidStent-1 and Group BraidStent-2. For example, how much of GlycomerTM 631 is combined with Poly-4-Hydroxybutyrate and how much of GlycomerTM 631 is combined with PGA?

Thank you for your helpful comment. This has been included in the manuscript.

“The ratio in the composition of the polymers of each stent was always kept constant in their manufacture: GlycomerTM 631 (54%), PGA and poly-4-hydroxybutyrate (46%).”

R.2.3. Please provide weight change data from protocol-I to protocol-II, BraidStent-2 group and BraidStent-SF group.

These data can be found in Table 1. T0: BraidStent-2 (0.027±0.004 mg) versus BraidStent-SF (0.044±0.005 mg), is 62% heavier.

R.2.4. The authors mentioned in Section 2 on the time points of the degradation as follows: “Four follow-ups were performed in each group: T0 – start of the study; T1–start of degradation under macroscopic control; T2–50% stent degradation; and T3–complete stent degradation.” This statement is not consistent with Figure 5. Please reconsider the expression of time points. Also, in Figure 5, what is the title for the y-axis?

Thank you for your right comment.

The y-axis is the days against the 4 follow-ups until the end of the study (T3, after complete stent degradation). This data has been included in the Figure 5. We consider that the graph is consistent, although we have detected that possibly our explanation of the follow-ups is not too appropriate and therefore leads to error. The graph shows the relationship between the days on which follow-ups occur for the five study groups. We believe that the definition of our follow-up points is what needs to be improved for a better understanding.

We have tried to clarify the wording for a better understanding: “Four follow-ups were performed in each group: T0 – start day of the study; T1–day of onset of macroscopic degradation; T2– day on which we macroscopically detected that the stent has degraded by 50%; and T3–complete stent degradation.” This has been included in the manuscript.

R.2.5. For protocol-III, what are the drug loadings for the BSFM1 and BSFM2 groups?

For BSFM1, MMC of 10 mg concentration was used, and the joint dilution with SF was 5 mg/mL with 10 coating cycles. For BSFM2, a pure MMC concentration of 70 mg was used and the joint SF dilution was 10 mg/mL, also with 10 dip coating cycles. The determination of the load is done by determination of the released MMC in the artificial urine (by HPLC-DAD), followed up until there is no more release of MMC in the urine. Our studies go up to 10 days, but in no case do we find MMC in artificial urine after 12h.

R.2.6. How was the in vitro MMC release was conducted and how were the MMC concentrations at various time points analyzed?

MMC concentrations are determined at follow-ups by HPLC-DAD. Urine from each follow-up is replaced and analyzed. MMC is released from the SF matrix due to solubility events of the SF matrix in the urinary medium. The permeability and release kinetics of the SF coating are related by the percentage of Beta-sheet structure. Increase the crystallinity (methanol dip) of the SF Beta-sheet decrease release rate and increased the duration of release.

Reviewer 3 Report

This manuscript is mainly about the assessment of the biodegradable ureteral stent coated with a silk fibroin matrix for the controlled release of mitomycin C as a chemotherapeutic drug. This surely can be important for treating upper urothelial carcinoma. In general, I recommend the acceptance of the manuscript after completely addressing the following concerns.

(1) Line 86~90, why only three polymers were considered? Three seems to be an extremely small number.

(2) Is it possible to combine Figure 2~4? Are these figures necessary? In addition, readers cannot judge the size of the sample solely from these figures.

(3) Line 142~157, I think a schematic figure on the possible structure of “ureteral stent coated with a silk fibroin matrix” may be helpful for readers.

(4) Is it possible to combine Figure 6~7? I actually gain little information from these figures.

(5) Are these coated ureteral stent stable in air? Will they peel off from ureteral stent?

Author Response

R.3.1. Line 86~90, why only three polymers were considered? Three seems to be an extremely small number.

We fully understand the reviewer's comment. This study is the result of a long line of research in which we have been evaluating and testing different synthetic polymers and copolymers for use in biodegradable ureteral stents. In this manuscript we have evaluated a new, very slow degrading polymer (poly-4-hydroxybutirate), compared to others better known from our previous studies such as PGA and Glycomer 631.

  1. de la Cruz JE, et al. Biodegradable ureteral stents: in vitro assessment of the degradation rates of braided synthetic polymers and copolymers. Am J Clin Exp Urol. 2022.
  2. Soria F, et al. Heparin coating in biodegradable ureteral stents does not decrease bacterial colonization-assessment in ureteral stricture endourological treatment in animal model. Transl Androl Urol. 2021.
  3. Soria F, et al. Comparative assessment of biodegradable-antireflux heparine coated ureteral stent: animal model study. BMC Urol. 2021.
  4. Soria F, et al. Experimental Assessment of New Generation of Ureteral Stents: Biodegradable and antireflux properties. J Endourol. 2020.

R.3.2. Is it possible to combine Figure 2~4? Are these figures necessary? In addition, readers cannot judge the size of the sample solely from these figures.

In accordance with your wise recommendations, we have deleted figure 4. Figure 2, a marking has been added so that readers can understand the dimensions of the stent fragment. We find this figure 2 very interesting as it allows readers to appreciate the SF coating in the stent. It also allows a comparison with Figure 3, which shows a BraidStent-SF in T2 with almost complete absence of the SF coating.

R.3.3. Line 142~157, I think a schematic figure on the possible structure of “ureteral stent coated with a silk fibroin matrix” may be helpful for readers.

Thank you for your comment. A schematic figure has been added to the manuscript.

R.3.4. Is it possible to combine Figure 6~7? I actually gain little information from these figures.

We certainly consider these two figures very important in our study. Figure 6 confirms the complete degradation of the stent in artificial urine and allows us to assess the small size of the degradation fragments, which, although not important at this stage of stent development, is important for clinical application, as these fragments must be very small in order not to cause obstruction of the upper urinary tract in patients.

Figure 7 is equally interesting as it allows visual confirmation that the MMC has been released into the urine from the stent. The MMC in contact with urine stains it magenta and the intensity of the colour is related to the free concentration in the artificial urine.

More information has been added in the figure caption of Figure 6 for better understanding.

R.3.5. Are these coated ureteral stent stable in air? Will they peel off from ureteral stent?

The stents were completely stable in room air, although storage should be in airtight packages as moisture can cause the coating and polymers to degrade more quickly. However, the MMC coating does need special care, as it must be packaged in a lightproof for preservation and transportation, to ensure its physical and chemical stability. MMC shows some light sensitivity, so reasonable steps to minimise light exposure should be taken. We use opaque airtight tubes for storage at temperatures below 28°C.

We have included a sentence in the manuscript to inform about the careful storage of the stents in M&M.

Round 2

Reviewer 1 Report

Thanks for sending us the revised version of your manuscript, which is now much improved and more suitable for publication. Nevertheless, I suggest you to add FTIR data (if possible) and this reference regarding drug release behavior: 

Iqbal H, Razzaq A, Uzair B, Ul Ain N, Sajjad S, Althobaiti NA, Albalawi AE, Menaa B, Haroon M, Khan M, Khan NU, Menaa F. Breast Cancer Inhibition by Biosynthesized Titanium Dioxide Nanoparticles Is Comparable to Free Doxorubicin but Appeared Safer in BALB/c Mice. Materials (Basel). 2021 Jun 8;14(12):3155. doi: 10.3390/ma14123155. PMID: 34201266; PMCID: PMC8229371.

best,

the reviewer

Author Response

R1.1.

We thank the reviewer for their careful reading of the manuscript and their constructive remarks.

-Unfortunately, we have not been able to perform FTIR in our study. All the material used was manufactured in the most aseptic way possible and always worked in a laminar flow cabinet. For future studies we will include an FTIR analysis, to rule out any materials that could be the source of contamination.

Thank you for your very valuable comment; we have already forwarded it to our Bioengineering Department to include FTIR in our methodology.

-We have included the recommended reference and a sentences in the manuscript (Discussion). [34].

All the best.

Reviewer 2 Report

(1)  R.2.3. Please provide weight change data from protocol-I to protocol-II, BraidStent-2 group and BraidStent-SF group.

Authors’ Responses: These data can be found in Table 1. T0: BraidStent-2 (0.027±0.004 mg) versus BraidStent-SF (0.044±0.005 mg), is 62% heavier.

Reviewer’s Comment: Table 1 in the revised manuscript is related to MMC release. Also, the revised table does not make sense.

(2)  R.2.5. For protocol-III, what are the drug loadings for the BSFM1 and BSFM2 groups?

Authors’ Responses: For BSFM1, MMC of 10 mg concentration was used, and the joint dilution with SF was 5 mg/mL with 10 coating cycles. For BSFM2, a pure MMC concentration of 70 mg was used and the joint SF dilution was 10 mg/mL, also with 10 dip coating cycles. The determination of the load is done by determination of the released MMC in the artificial urine (by HPLC-DAD), followed up until there is no more release of MMC in the urine. Our studies go up to 10 days, but in no case do we find MMC in artificial urine after 12h.

Reviewer’s Comment: Please reword the responses and include them in Protocol III statement in Section 3. Results.

(3)  R.2.6. How was the in vitro MMC release was conducted and how were the MMC concentrations at various time points analyzed?

Authors’ Responses: MMC concentrations are determined at follow-ups by HPLC-DAD. Urine from each follow-up is replaced and analyzed. MMC is released from the SF matrix due to solubility events of the SF matrix in the urinary medium. The permeability and release kinetics of the SF coating are related by the percentage of Beta-sheet structure. Increase the crystallinity (methanol dip) of the SF Beta-sheet decrease release rate and increased the duration of release.

Reviewer’s Comment: Please try to include the statement and the HPLC methods in Section 2. Materials and Methods. I understood that the authors had one sentence in Protocol III in Section 2. Materials and Methods. There are more to provide in HPLC methods, including mobile phases (compositions), retention time, column temperature, flow rate, injection volume, UV-vis detection wavelength, and etc. Please try to separate this section out from the previous version. Please make it stand alone in Materials and Methods section.

Author Response

We thank the reviewer for their careful reading of the manuscript and their constructive remarks.

R.2.3.

You are absolutely right. We made the big mistake of revising the manuscript on the advice of the reviewers and once submitted, we realised that the editor had also requested changes to the manuscript. As the replies to the reviewers had already been sent, there is therefore an inconsistency with the final manuscript. The editor asked us to change tables 1 to 3 to figures to improve for better understanding. Specifically, figure 4 corresponds to the results of the stent weights of the 5 groups. Figure 8 corresponds to the assessment of stent thickness at baseline.

R.2.5.

Following your advice, we have included the following sentences in Protocol III Results.

“Both stents groups underwent the same number of coating cycles, 10, although the initial concentration of MMC in BraidStent-SF-MMC1 was 10 mg compared to 70 mg in BraidStent-SF-MMC2. As well as the joint dilution of SF and MMC was 5 mg/ml and 10 mg/ml respectively. The kinetics of MMC in both groups was fully released within the first 12h, with no analytical evidence of MMC afterwards (Figure10). We found statistically significant differences in the concentration of MMC released at 12 h between the groups, with a higher urine concentration of MMC in the BraidStent-SF-MMC2 group (Table 1). Statistically significant differences were also found within each group between 6 and 12 h. None of the samples were positive for nitrites on urinalysis.”

R.2.6.

Following your fine advice, we have included a new section in M&M. Protocol III.

M&M. Protocol III.

MMC assessment.

MMC concentrations were assessed every six hours until there was no analytical evidence of its detection. Urine from each follow-up was replaced and analysed. MMC is released from the SF matrix due to solubility events of the SF matrix in the urinary environment. The permeability and release kinetics of the MMC depends on the SF coating and relates to the percentage of beta-sheet structure. Increasing the crystallinity (methanol immersion) of the SF beta sheet decreases the release rate and increases the release duration.

The HPLC-DAD method was an isocratic method with a mobile phase of acetonitrile: ultrapure water 80:20 (v/v) at a flow rate of 1 mL/min and separation was performed on LUNA C18 250 mm, 4.6 mm, 5 µm column at 30 ºC temperature. 10µL of sample was injected. The MMC was detected with a diode array detector (DAD) at 365 nm (1260 Infinity II Prime LC System, Agilent Technologies).

Round 3

Reviewer 2 Report

Thank you for addressing my questions. I have no further questions.

Author Response

Dear Reviewer.

 We appreciate the time and effort that you dedicated to providing feedback on our
manuscript and are grateful for the insightful comments on and valuable improvements to our scientific paper.